# CONTINUAL LEARNING IN LOW-COHERENCE SUBSPACE: A STRATEGY TO MITIGATE LEARNING CAPACITY DEGRADATION

## ABSTRACT

Methods using gradient orthogonal projection, an efficient strategy in continual learning, have achieved promising success in mitigating catastrophic forgetting. However, these methods often suffer from the learning capacity degradation problem following the increasing number of tasks. To address this problem, we propose to learn new tasks in low-coherence subspaces rather than orthogonal subspaces. Specifically, we construct a unified cost function involving regular DNN parameters and gradient projections on the Oblique manifold. We finally develop a gradient descent algorithm on a smooth manifold to jointly minimize the cost function and minimize both the inter-task and the intra-task coherence. Numerical experimental results show that the proposed method has prominent advantages in maintaining the learning capacity when tasks are increased, especially on a large number of tasks, compared with baselines. Li & Lin (2018)

## 1 INTRODUCTION

**D**eep **N**eural **N**etworks (DNNs) have achieved promising performance on many tasks. However, they lack the ability for continual learning, i.e., they suffer from catastrophic forgetting French (1999) when learning sequential tasks, where catastrophic forgetting is a phenomenon of new knowledge interfering with old knowledge. Research on continual learning, also known as incremental learning Aljundi et al. (2018a); Chaudhry et al. (2018a); Chen & Liu (2018); Aljundi et al. (2017), and sequential learning Aljundi et al. (2018b); McCloskey & Cohen (1989), aims to find effective algorithms that enable DNNs to simultaneously achieve plasticity and stability, i.e., to achieve both high learning capacity and high memory capacity.

Various methods have been proposed to avoid or mitigate the catastrophic forgetting De Lange et al. (2019), either by replaying training samples Rolnick et al. (2019); Ayub & Wagner (2020); Saha et al. (2021), or reducing mutual interference of model parameters, features or model architectures between different tasks Zenke et al. (2017); Mallya & Lazebnik (2018); Wang et al. (2021). Among these methods, **G**radient **O**rthogonal **P**rojection (GOP) Chaudhry et al. (2020); Zeng et al. (2019); Farajtabar et al. (2020); Li et al. (2021) is an efficient continual learning strategy that advocates projecting gradients with the orthogonal projector to prevent the knowledge interference between tasks. GOP-based methods have achieved encouraging results in mitigating catastrophic forgetting. However, from Fig. 1, we observe that these methods suffer from the learning capacity degradation problem: their learning capacity is gradually degraded as the number of tasks increases and eventually becomes unlearnable. Specifically, when learning multiple tasks, e.g., more than 30 tasks in Fig. 1, their performance on new tasks dramatically decreases. These results suggest that the GOP-based methods focus on the stability and somewhat ignore the plasticity. Ignoring the plasticity may limit the task learning capacity of models, i.e., the number of tasks that a model can learn without forgetting.

To address this issue, we propose to learn new tasks in low-coherence subspaces rather than orthogonal subspaces. Specifically, Low-coherence projectors are utilized for each layer to project features and gradients into low-coherence subspaces. To achieve this, we construct a unified cost function to find projectors and develop a gradient descent algorithm on the Oblique manifold to jointly minimize inter-task coherence and intra-task coherence. Minimizing the inter-task coherence can reduce the mutual interference between tasks, and minimizing the intra-task coherence can enhance the model's expressive power. Restricting projectors on the Oblique manifold can avoid the scale ambiguity

Aharon et al. (2006); Wei et al. (2017), i.e., preventing the parameters of the projector from being extremely large or extremely small.

The main contributions of this work are summarized as follows. First, to address the learning capacity degradation problem of GOP, we propose a novel method, namely, **L**ow-**c**oherence **S**ubspace **P**rojection (LcSP), that replaces the orthogonal projectors with the low-coherence gradient projectors, allowing the DNN to maintain both plasticity and stability. Additionally, our work observes that the GOP models with **B**atch **N**ormalization (BN) Ioffe & Szegedy (2015) layers could cause catastrophic forgetting. This paper proposes two strategies in LcSP to solve this problem, i.e., replacing BN with **G**roup **N**ormalization (GN) Wu & He (2018) or learning specific BN for each task.

## 2 RELATED WORK

In this section, we briefly review some existing works of continual learning and the GOP based methods.

**Replay-based Strategy**   The basic idea of this type of approach is to use limited memory to store small amounts of data (e.g., raw samples) from previous tasks, called episodic memory, and to replay them when training a new task. Some of the existing works focused on selecting a subset of raw samples from the previous tasks Rolnick et al. (2019); Isele & Cosgun (2018); Chaudhry et al. (2019); Zhang et al. (2020). In contrast, others concentrated on training a generative model to synthesize new data that can substitute for the old data Shin et al. (2017); Van de Ven & Tolias (2018); Lavda et al. (2018); Ramapuram et al. (2020).

**Regularization-based Strategy**   This strategy prevents catastrophic forgetting by introducing a regularization term in the loss function to penalize the changes in the network parameters. Existing works can be divided into data-focused, and prior-focused methods De Lange et al. (2021). The Data-focused methods take the previous model as the teacher and the current model as the student, transferring the knowledge from the teacher model to the student model through knowledge distillation. Typical methods include LwF Li & Hoiem (2017), LFL Jung et al. (2016), EBLL Rannen et al. (2017), DMC Zhang et al. (2020) and GD-WILD Lee et al. (2019). The prior-focused methods estimate a distribution over the model parameters, assigning an importance score to each parameter and penalizing the changes in significant parameters during learning. Relevant works include SI Zenke et al. (2017), EWC Kirkpatrick et al. (2017), RWalk Chaudhry et al. (2018a), AGS-CL Jung et al. (2020) and IMM Lee et al. (2017).

**Parameter Isolation-based Strategy**   This strategy considers dynamically modifying the network architecture by pruning, parameter mask, or expansion to greatly or even completely reduce catastrophic forgetting. Existing works can be roughly divided into two categories. One is dedicated to isolating separate sub-networks for each task from a large network through pruning techniques and parameter mask, including PackNet Mallya & Lazebnik (2018), PathNet Fernando et al. (2017), HAT Serra et al. (2018) and Piggyback Mallya et al. (2018). Another class of methods dynamically expands the network architecture, increasing the number of neurons or sub-network branches, to break the limits of expressive capacity (Rusu et al., 2016; Aljundi et al., 2017; Xu & Zhu, 2018; Rosenfeld & Tsotsos, 2018). However, as the number of tasks growing, this approach also complicates the network architecture and increases the computation and memory consumption.

**Gradient Orthogonal Projection-based Strategy**   Methods based on GOP strategies, which reduce catastrophic forgetting by projecting gradient or features with orthogonal projectors, have been shown to be effective in continual learning with encouraging results Farajtabar et al. (2020); Zeng et al. (2019); Saha et al. (2021); Wang et al. (2021); Chaudhry et al. (2020). According to the different ways of finding the projector, we can further divide the existing works into **C**ontext **O**rthogonal **P**rojection (COP) and **S**ubspace **O**rthogonal **P**rojection (SOP). Methods based on COP, such as OWM Zeng et al. (2019), Adam-NSCL Wang et al. (2021), and GPM Saha et al. (2021), always rely on the context of previous tasks to build projectors. In contrast to COP, SOP-based methods such as ORTHOG-SUBSPACE Chaudhry et al. (2020) use hand-crafted, task-specific orthogonal projectors and yield competitive results.

This paper proposes a novel approach to continual learning called LcSP. Compared with other methods based on GOP, LcSP trains the network on the low-coherence subspaces to balance plasticity and stability and overcomes the learning capacity degradation problem, which significantly decreases the performance of GOP methods with the increasing number of tasks.

## 3    CONTINUAL LEARNING SETUP

This work adopts the Task-Incremental Learning (TIL) setting, where multiple tasks are learned sequentially. Let us assume that there are $T$ tasks, denoted by $\mathcal{T}_t$ for $t = 1, \ldots, T$ with its training data $\mathcal{D}_t = \{(x_i, y_i, \tau_t)_{i=1}^{N_t}\}$. Here, the data $(x_i, y_i) \in \mathcal{X} \times \mathcal{Y}_t$ is assumed to be drawn from some independently and identically distributed random variable, and $\tau_t \in \mathcal{T}$ denotes the task identifier. In the TIL setting, the data $\mathcal{D}_t$ can be accessed if and only if task $\mathcal{T}_t$ arrives. When episodic memory is adopted, a limited number of data samples drawn from old tasks can be stored in the replay buffer $\mathcal{M}$ so that $\mathcal{D}_t \cup \mathcal{M}$ can be used for training when task $\mathcal{T}_t$ arrives.

Assuming that a network $f$ parameterized with $\Phi = \{\theta, \varphi\}$ consists of two parts, where $\theta$ denotes the parameters of the backbone network and $\varphi$ denotes the parameters of the classifier. Let $f(x; \theta) :$ $\mathcal{X} \times \mathcal{T} \to \mathcal{H}$ denote the backbone network parameterized with $\theta = \{W_l\}_{l=1}^L$, which encodes the data samples $x$ into feature vector. Let $f(x; \varphi) : \mathcal{H} \to \mathcal{Y}$ denote the classifier parameterized with $\varphi = w$ which returns the classification result of the feature vector obtained by $f(x; \theta)$. The goal of TIL is to learn $T$ tasks sequentially with the network $f$ and finally achieve the optimal loss on all tasks.

**Evaluation Metrics**   Once the training on all tasks is finished, we evaluate the performance of algorithm by calculating the average accuracy $\mathcal{A}$ and forgetting $\mathcal{F}$ Chaudhry et al. (2020) of the network on the T tasks $\{\mathcal{T}_1, ..., \mathcal{T}_T\}$. Suppose all tasks come sequentially, let $Acc_{i,j}$ denote the test accuracy of the network on task $\mathcal{T}_i$ after learning task $\mathcal{T}_j$, where $i \leq j$. The average accuracy is defined as

$$\mathcal{A} = \frac{1}{T} \sum_{i=1}^{T} Acc_{i,T}, \tag{1}$$

and the forgetting is defined as

$$\mathcal{F} = \frac{1}{T-1} \sum_{i=1}^{T-1} \max_{j \in \{i, ..., T-1\}} (Acc_{i,j} - Acc_{i,T}). \tag{2}$$

## 4    CONTINUAL LEARNING IN LOW-COHERENCE SUBSPACES

In this section, we first introduce how to find task-specific, low-coherence projectors for LcSP on the Oblique manifold. We then describe how to use it in a specific DNN architecture to project features and gradients. Finally, we analyze the factors that enable LcSP to maintain plasticity and stability.

### 4.1    CONSTRUCTING LOW-COHERENCE PROJECTORS ON OBLIQUE MANIFOLD

Here, we first introduce the concept of coherence metric. The coherence metric is usually used in compressed sensing and sparse signal recovery to describe the correlation of the columns of a measurement matrix Candes et al. (2011); Candes & Romberg (2007). Formally, the coherence of a matrix $\boldsymbol{M}$ is defined as

$$\mu(\boldsymbol{M}, \boldsymbol{N}) = \begin{cases} \max_{j<k} \frac{|\langle \boldsymbol{M}_j, \boldsymbol{M}_k \rangle|}{\|\boldsymbol{M}_j\|_2 \|\boldsymbol{M}_k\|_2}, & \boldsymbol{M} = \boldsymbol{N} \\ \\ \max_{i,j} \frac{|\langle \boldsymbol{M}_i, \boldsymbol{N}_j \rangle|}{\|\boldsymbol{M}_i\|_2 \|\boldsymbol{N}_j\|_2}, & \boldsymbol{M} \neq \boldsymbol{N} \end{cases} . \tag{3}$$

where $\boldsymbol{M}_j$ and $\boldsymbol{M}_k$ denote the column vectors of matrix $\boldsymbol{M}$. Without causing confusion, we use $\mu(\boldsymbol{M})$ denote $\mu(\boldsymbol{M}, \boldsymbol{M})$. To measure the coherence between different projectors, we introduce the

Babel function Li & Lin (2018), measuring the maximum total coherence between a fixed atom and a collection of other atoms in a dictionary, which can be described as follows.

$$\mathbf{B}(\boldsymbol{M}) = \max_{\Lambda, |\Lambda| = M} \max_{i \notin \Lambda} \sum_{j \in \Lambda} \frac{|\langle \boldsymbol{M}_i, \boldsymbol{M}_j \rangle|}{\|\boldsymbol{M}_i\| \, \|\boldsymbol{M}_j\|} \tag{4}$$

With the concept of a coherence metric in mind, we then introduce the main optimization objective in finding projectors. Specifically, suppose that the DNN has learned the task $\mathcal{T}_1, \mathcal{T}_2, ..., \mathcal{T}_{t-1}$ in the subspace $\mathcal{S}_1, \mathcal{S}_2, ..., \mathcal{S}_{t-1}$, respectively, and $\boldsymbol{P}_1, \boldsymbol{P}_2, .., \boldsymbol{P}_{t-1}$ denote the projectors of all previous tasks. When learning task $\mathcal{T}_t$, we project features and gradients into a $d_t$-dimensional low-coherence subspace $\mathcal{S}_t$ with projector $\boldsymbol{P}_t$ so that the LcSP can prevent catastrophic forgetting. The projector $\boldsymbol{P}_t$ can be found by optimizing

$$\begin{aligned} &\arg\min \mathbf{B}(\boldsymbol{P}_t), \\ &\text{s.t.} \quad \boldsymbol{P}_t \in \mathbb{R}^{m \times m}, \quad \operatorname{rank}(\boldsymbol{P}_t) = d_t. \end{aligned} \tag{5}$$

Two considerations need to be taken in solving Eq. (5), i.e., considering the rank constraint and avoiding the entries of $\boldsymbol{P}_t$ being extremely large or extremely small. With these considerations in mind, we can rephrase the rank and scale constrained problem as a problem on the Riemannian manifold, more specifically on the Oblique manifold $\mathcal{OM}(m, d_t)$, by setting $\boldsymbol{P}_t = \boldsymbol{O}_t \boldsymbol{O}_t^\top$ and normalizing the columns of $\boldsymbol{O}_t$, i.e, $\operatorname{diag}(\boldsymbol{O}_t^\top \boldsymbol{O}_t) = \boldsymbol{I}_n$, where $\operatorname{diag}(\cdot)$ represents the diagonal matrix and $\boldsymbol{I}_n$ is the $n \times n$ identity matrix. With these settings, the new cost function $\mathbf{J}(\cdot)$ and optimization problem can be described as follows:

$$\mathbf{J}(\boldsymbol{O}_t) = \begin{cases} \lambda \cdot \mathbf{B}(\boldsymbol{O}_t \boldsymbol{O}_t^\top) + \gamma \cdot \mu(\boldsymbol{O}_t \boldsymbol{O}_t^\top), & t > 1 \\ \mu(\boldsymbol{O}_t \boldsymbol{O}_t^\top), & t = 1 \end{cases},$$
$$\boldsymbol{O}_t = \arg\min \mathbf{J}(O_t), \quad \text{s.t.} \quad \boldsymbol{O}_t \in \mathcal{OB}(m, d_t). \tag{6}$$

In the cost function $\mathbf{J}(\boldsymbol{O}_t)$, we further divided the optimization objective into inter-task $\mathbf{B}(\boldsymbol{O}_t \boldsymbol{O}_t^\top)$ and intra-task $\mu(\boldsymbol{O}_t \boldsymbol{O}_t^\top)$, and utilize parameters $\lambda$ and $\gamma$ to provide a trade-off between them. Here, intra-task coherence is optimized to maintain the full rank of $\boldsymbol{O}_t$. Meeting the full-rank constraint helps to balance plasticity and stability in the case of increasing tasks. Relevant ablation studies and numerical analyses are given in the appendix.

Optimization on the Oblique manifold, i.e., the solution lies on the Oblique manifold, is a well-established area of research Absil et al. (2009); Absil & Gallivan (2006); Selvan et al. (2012). Here, we briefly summarize the main steps of the optimization process. Formally, the Oblique manifold $\mathcal{OM}(n, p)$ is defined as

$$\mathcal{OM}(n, p) \triangleq \{\boldsymbol{X} \in \mathbb{R}^{n \times p} : \operatorname{diag}(\boldsymbol{X}^\top \boldsymbol{X}) = \boldsymbol{I}_p\}, \tag{7}$$

representing the set of all $n \times p$ matrices with normalized columns. $\mathcal{OM}$ can also be considered as an embedded Riemannian manifold of $\mathbb{R}^{n \times p}$, endowed with the canonical inner product

$$\langle \boldsymbol{X}_1, \boldsymbol{X}_2 \rangle = \operatorname{trace}\left(\boldsymbol{X}_1^\top \boldsymbol{X}_2\right), \tag{8}$$

where $\operatorname{trace}(\cdot)$ represents the sum of the diagonal elements of the given matrix. For a given point $\boldsymbol{X}$ on $\mathcal{OM}$, the tangent space at $\boldsymbol{X}$, denoted by $T_{\boldsymbol{X}}\mathcal{OM}$, is defined as

$$T_{\boldsymbol{X}}\mathcal{OM}(n, p) = \{\boldsymbol{U} \in \mathbb{R}^{n \times p} : \operatorname{diag}(\boldsymbol{X}^\top \boldsymbol{U}) = 0\}. \tag{9}$$

Further, the tangent space projector $\mathbf{P}_{\boldsymbol{X}}$ at $\boldsymbol{X}$ which projects $\boldsymbol{H} \in \mathbb{R}^{n \times p}$ into $T_{\boldsymbol{X}}\mathcal{OM}$, is represented as

$$\mathbf{P}_{\boldsymbol{X}}(\boldsymbol{H}) = \boldsymbol{H} - \boldsymbol{X} \operatorname{ddiag}\left(\boldsymbol{X}^\top \boldsymbol{H}\right), \tag{10}$$

where $\operatorname{ddiag}$ sets all off-diagonal entries of a matrix to zero. When optimizing on $\mathcal{OM}$, the $k$th iterate $\boldsymbol{X}_k$ must move along a descent curve on $\mathcal{OM}$ for the cost function, such that the next iterate $\boldsymbol{X}_{k+1}$ will be fixed on the manifold. This is achieved by the retraction

$$\mathbf{R}_{\boldsymbol{X}_k}(\boldsymbol{U}) = \operatorname{normalize}(\boldsymbol{X}_k + \boldsymbol{U}), \tag{11}$$

where $\operatorname{normalize}$ scales each column of the input matrix to have unit length. Finally, with this knowledge, we can extend the gradient descent algorithm to solve any unconstrained optimization problems on $\mathcal{OM}$, which can be summarized as

$$\begin{aligned} \boldsymbol{U} &= \mathbf{P}_{\boldsymbol{X}_k}(\nabla_{\boldsymbol{X}_k}\mathbf{J}), \\ \boldsymbol{X}_{k+1} &= \mathbf{R}_{\boldsymbol{X}_k}(-\alpha \boldsymbol{U}), \end{aligned} \tag{12}$$

where $\nabla_{\boldsymbol{X}_k} \mathbf{J}$ denotes the Euclidean gradient at the $k$th iterate and $\alpha$ is the step size. Finally, our algorithm for finding $\boldsymbol{O}_t$ in $\mathcal{OM}$ to task $\mathcal{T}_t$ is summarized in Algorithm 1.

---

**Algorithm 1** Construct the $\boldsymbol{O}_t$ on $\mathcal{OM}$ for Task $\mathcal{T}_t$

---

**Input:** $\boldsymbol{O}_1, ..., \boldsymbol{O}_{t-1}$
**Output:** $\boldsymbol{O}_t$
1: $\mathbf{R}_{\boldsymbol{X}}(\boldsymbol{U}) := \text{normalize}(\boldsymbol{X} + \boldsymbol{U})$         ▷ normalize scales each column of the input matrix to have norm 1
2: $\boldsymbol{X}_0 \leftarrow$ random initialization on $\mathcal{OM}$
3: $k \leftarrow 0$
4: Set tolerance error $0 \leq \mathcal{E} \ll 1$
5: **while** True **do**
6:      $\boldsymbol{G} \leftarrow \nabla f(\boldsymbol{X}_k)$         ▷ Calculate Euclidean Gradient
7:      $\boldsymbol{U} \leftarrow \boldsymbol{G} - \boldsymbol{X}_k \, \text{ddiag}(\boldsymbol{X}_k^\top \boldsymbol{G})$         ▷ Calculate Riemann Gradient
8:      **if** $\|\boldsymbol{U}\| \leq \mathcal{E}$ **then**
9:         break
10:      **end if**
11:      $\alpha \in (0, 0.5), \quad \beta \in (0, 1)$
12:      $t \leftarrow 1$         ▷ Initial step size
13:      **while** $\mathbf{J}(\mathbf{R}_{\boldsymbol{X}_k}(-t \cdot \boldsymbol{U})) > \mathbf{J}(\boldsymbol{X}_k) - \alpha \cdot t \cdot \|\boldsymbol{U}\|_2^2$ **do**         ▷ Searching the step size for calculating next iterate
14:         $t \leftarrow \beta \cdot t$
15:      **end while**
16:      $\boldsymbol{X}_{k+1} \leftarrow \mathbf{R}_{\boldsymbol{X}_k}(-t \cdot \boldsymbol{U})$         ▷ Updating the next iterate $\boldsymbol{X}_{k+1}$ and fixing it on manifold
17:      $k \leftarrow k + 1$
18: **end while**
19: $\boldsymbol{O}_t = \boldsymbol{X}_k$
20: **return** $\boldsymbol{O}_t$

---

### 4.2 THE APPLICATION OF LOW-COHERENCE PROJECTORS IN DNNS

With the LcSP at hand, the following introduces some technical details of applying LcSP in DNNs. When learning task $\mathcal{T}_t$, LcSP first constructs task-specific projector $\boldsymbol{P}_t^l$ for each layer before training, and freezes them during training. These projectors are used to project the features and gradients, ensuring that the DNN learns in the low-coherence subspace. Specifically, suppose that a network $f$ with $L$ linear layers is used as DNN architecture, let $\boldsymbol{W}_t^l$, $\boldsymbol{x}_t^l$, $\boldsymbol{z}_t^l$, $\sigma^l$, and $\boldsymbol{P}_t^l$ denote the model parameters, the input features, the output features, the activation function, and the introduced low-coherence projector in layer $l \in \{1, ..., L\}$, respectively. LcSP introduces $\boldsymbol{P}_t^l$ immediately after $\boldsymbol{W}_t^l$ such that the pre-activation features are projected into the subspace, i.e.,

$$\boldsymbol{z}_t^l = (\boldsymbol{x}_t^l \boldsymbol{W}_t^l) \boldsymbol{P}_t^l,$$
$$\boldsymbol{x}_t^{l+1} = \sigma^l(\boldsymbol{z}_t^l). \tag{13}$$

According to the chain rule of derivation, the gradients at $\boldsymbol{W}_t^l$ will also be multiplied with $\boldsymbol{P}_l^t$ in backpropagation, as follows

$$
\begin{aligned}
\frac{\partial \mathcal{L}}{\partial (\boldsymbol{W}_t^l)_{(i,:)}} &= \frac{\partial \mathcal{L}}{\partial \boldsymbol{z}_t^l} \frac{\partial \boldsymbol{z}_t^l}{\partial (\boldsymbol{W}_t^l)_{(i,:)}} \\
&= \frac{\partial \mathcal{L}}{\boldsymbol{z}_t^L} \prod_{k=l}^{L-1} \frac{\partial \boldsymbol{z}_t^{k+1}}{\partial \boldsymbol{z}_t^k} \cdot (\boldsymbol{x}_t^l)_i \cdot \boldsymbol{P}_t^l,
\end{aligned}
\tag{14}
$$

where $(\boldsymbol{W}_t^l)_{(i,:)}$ represents the $i$th row of $\boldsymbol{W}_t^l$ and $(\boldsymbol{x}_t^l)_i$ is the $i$th element of $\boldsymbol{x}_t^l$. In **C**onvolutional **N**eural **N**etworks (CNNs), the input and the output typically represent the image features and have more than two dimensions, e.g., input channel, output channel, height, and width. In this case, we reshape $\boldsymbol{z}^l \in \mathbb{R}^{c_{\text{out}} \times (c_{in} \cdot h \cdot w)}$ to $\boldsymbol{z}^l \in \mathbb{R}^{(c_{in} \cdot h \cdot w) \times c_{\text{out}}}$ and align the dimension of projector with the output channel so that $\boldsymbol{P}_t^l \in \mathbb{R}^{c_{\text{out}} \times c_{\text{out}}}$. After the projection, we recover the shape of $\boldsymbol{z}_t^l$ so that it can be used as input for the next layer.

**Overcoming the Catastrophic Forgetting in BN based models**    BN is a widely used module in DNNs to make training of DNNs faster and more stable through normalization of the layers' features by re-centering and re-scaling Ioffe & Szegedy (2015). However, re-centering and re-scaling of the layers' features changes the data distribution (e.g., the mean and the variance) of features of previous tasks, which often leads to the catastrophic forgetting of LcSP. For example, when learning the new

task $\mathcal{T}_t$, $\boldsymbol{W}_t^l$ may not work for $\mathcal{T}_t$ due to the change in data distribution caused by BN. To solve this problem, we propose two strategies in LcSP: **the strategy (1)** learning specific BN for each task, or **the strategy (2)** using GN instead of BN. We verify the effectiveness of these two strategies in experiments and compare their performance in §5.

### 4.3 METHOD ANALYSIS

In this section, we provide analysis on plasticity and stability of LcSP.

**Stability Analysis** Let $\theta = \{\boldsymbol{W}_t^l\}_{l=1}^L$ denote the parameter set of $f$; $\Delta\theta = \{\Delta\boldsymbol{W}_t^1, \ldots, \Delta\boldsymbol{W}_t^L\}$ denote set of variation values of parameters after learning task $\mathcal{T}_t$; $\boldsymbol{P}_t = \{\boldsymbol{P}_t^l\}_{l=1}^L$ denote the projectors set obtained by Algorithm 1; $\boldsymbol{x}_{q,t}^l$ and $\boldsymbol{z}_{q,t}^l$ denote the input and ouput when feeding the data of task $\mathcal{T}_q$ ($q \leq t$) into the network $f$, which has been optimized in learning task $\mathcal{T}_t$.

**Lemma 1.** *Assume that $f$ is fed the data of task $\mathcal{T}_t$ ($q < t$), then $f$ can effectively overcomes catastrophic forgetting if*

$$\boldsymbol{z}_{q,q}^l \approx \boldsymbol{z}_{q,t}^l, \quad \forall q \leq t \tag{15}$$

*holds for $l \in \{1, 2, ..., L\}$.*

Lemma 1 suggests that $f$ can overcome catastrophic forgetting if the output of $f$ to previous tasks is invariant. In the following, we prove that LcSP achieves approximate invariance to the output of previous tasks.

***Proof.*** *Suppose $q = t - 1$. When $l = 1$, $\boldsymbol{x}_{q,t}^l = \boldsymbol{x}_{q,q}^l$. Then*

$$\begin{aligned}
\boldsymbol{z}_{q,t}^l &= \boldsymbol{x}_{q,t}^l (\boldsymbol{W}_q^l + \Delta\boldsymbol{W}_t^l)\boldsymbol{P}_q^l \\
&= \boldsymbol{x}_{q,t}^l \boldsymbol{W}_q^l P_q^l + \boldsymbol{x}_{q,t}^l \Delta\boldsymbol{W}_t^l \boldsymbol{P}_q^l \\
&= \boldsymbol{z}_{q,q}^l + \boldsymbol{x}_{q,t}^l \Delta\boldsymbol{W}_t^l \boldsymbol{P}_q^l.
\end{aligned} \tag{16}$$

*Let $g_t^l$ denote the gradient when training the network on task $\mathcal{T}_t$. In backpropagation, $\Delta\boldsymbol{W}_t^l = g_t^l \boldsymbol{P}_t^l$. Then $\boldsymbol{x}_{q,t}^l \Delta\boldsymbol{W}_t^l \boldsymbol{P}_q^l = \boldsymbol{x}_{q,t}^l g_t^l \boldsymbol{P}_t^l \boldsymbol{P}_q^l$. If the inter-task coherence $\mu(\boldsymbol{P}_t^l, \boldsymbol{P}_q^l) \approx 0$, then $\boldsymbol{P}_t^l \boldsymbol{P}_q^l \approx \boldsymbol{0}$. Projectors satisfying this condition can be found by Algorithm 1. We can prove that $\boldsymbol{z}_{q,q}^l \approx \boldsymbol{z}_{q,t}^l$ holds for all layers by repeating the above process. This proof can also be generalized to any previous task $\mathcal{T}_q$.*

**Plasticity Analysis** Let $\tilde{g}_t^l = g_t^l \boldsymbol{P}_t^l$ denote the projected gradient at $\boldsymbol{W}_t^l$. $f$ can achieve optimal loss on task $\mathcal{T}_t$ if $\langle g_t^l, \tilde{g}_t^l \rangle > 0$ holds for each $l \in \{1, \ldots, L\}$, where $\langle \cdot, \cdot \rangle$ represents the inner product. Here, we prove that $\langle g_t^l, \tilde{g}_t^l \rangle > 0$ holds for each $l \in \{1, \ldots, L\}$.

***Proof.*** *Let $\tilde{g}_t^l = g_t^l \boldsymbol{P}_t^l$ denote the projected gradient, we have*

$$\begin{aligned}
\langle g_t^l, \tilde{g}_t^l \rangle &= g_t \tilde{g}_t^{l\top} = g_t^l \boldsymbol{O}_t^l \boldsymbol{O}_t^{l\top} g_t^{l\top} \\
&= \langle g_t^l \boldsymbol{O}_t^l, g_t^l \boldsymbol{O}_t^l \rangle = \|g_t^l \boldsymbol{O}_t^l\| > 0.
\end{aligned} \tag{17}$$

*Note that $\|g_t^l \boldsymbol{O}_t^l\|$ is always positive unless $g_t^l \boldsymbol{O}_t^l$ is $\boldsymbol{0}$. This result is easy to generalize to each layer.*

## 5 EXPERIMENTS

In this section, we evaluate our approach on several popular continual learning benchmarks and compare LcSP with previous state-of-the-art methods. The result of accuracy and forgetting demonstrate the effectiveness of our LcSP, especially when the number of tasks is large.

### 5.1 BENCHMARKS

**Benchmarks for Learning 20 Tasks** We conducted this experiment on four image classification datasets: Permuted MNIST, Rotated MNIST, Split CIFAR100 and Split miniImageNet. Permuted MNIST is constructed by randomly rearranging MNIST LeCun (1998) image pixels, using different seeds for different tasks. Rotated MNIST is constructed by rotating the MNIST image at a certain

Table 1: The average accuracy and forgetting results of the proposed LcSP and baselines. **Memory** denotes whether the method is trained using a replay strategy with episodic memory.

| Methods | Memory | Permuted MNIST | | Rotated MNIST | |
| --- | --- | --- | --- | --- | --- |
| | | Accuracy(%) | Forgetting | Accuracy(%) | Forgetting |
| EWC Kirkpatrick et al. (2017) | ✗ | 68.4(±0.76) | 0.25(±0.01) | 43.6(±0.81) | 0.53(±0.01) |
| AGEM Chaudhry et al. (2018b) | ✓ | 78.3(±0.42) | 0.15(±0.01) | 60.5(±1.77) | 0.36(±0.01) |
| ER-RING Chaudhry et al. (2019) | ✓ | 79.5(±0.31) | 0.12(±0.01) | 70.9(±0.38) | 0.24(±0.01) |
| KCL Derakhshani et al. (2021) | ✗ | 85.5(±0.78) | **0.02**(±**0.00**) | 81.8(±0.60) | **0.01**(±**0.00**) |
| OWM Zeng et al. (2019) | ✗ | 46.35 | 0.01 | 44.78 | 0.01 |
| Adam-NSCL Wang et al. (2021) | ✗ | 26.44 | 0.70 | 42.73 | 0.56 |
| GPM Saha et al. (2021) | ✗ | 80.54 | 0.16 | 80.34 | 0.16 |
| ORTHOG-SUBSPACE Chaudhry et al. (2020) | ✗ | 86.6(±0.91) | 0.04(±0.01) | 80.1(±0.95) | 0.14(±0.01) |
| **LcSP** | ✗ | **92.2**(±**0.65**) | 0.05(±0.01) | **86.6**(±**0.96**) | 0.08(±0.01) |

| Methods | Memory | Split CIFAR100 | | Split miniImageNet | |
| --- | --- | --- | --- | --- | --- |
| | | Accuracy(%) | Forgetting | Accuracy(%) | Forgetting |
| EWC Kirkpatrick et al. (2017) | ✗ | 43.2(±2.77) | 0.26(±0.02) | 34.8(±2.34) | 0.24(±0.04) |
| AGEM Chaudhry et al. (2018b) | ✓ | 51.3(±3.49) | 0.18(±0.03) | 42.3(±1.42) | 0.17(±0.01) |
| ER-RINGChaudhry et al. (2019) | ✓ | 59.6(±1.19) | 0.14(±0.01) | 49.8(±2.92) | 0.12(±0.01) |
| HAT Serra et al. (2018) | ✗ | 72.06 | **0.00** | 59.78 | 0.03 |
| KCL Derakhshani et al. (2021) | ✗ | 62.7(±0.89) | 0.06(±0.01) | 53.3(±0.57) | 0.04(±0.00) |
| OWM Zeng et al. (2019) | ✗ | 50.94 | 0.30 | - | - |
| ORTHOG-SUBSPACE Chaudhry et al. (2020) | ✓ | 64.3(±0.59) | 0.07(±0.01) | 51.4(±1.44) | 0.10(±0.01) |
| GPM Saha et al. (2021) | ✗ | 72.48 | **0.00** | 60.41 | **0.00** |
| Adam-NSCL Wang et al. (2021) | ✗ | **75.95** | 0.04 | 63.27 | 0.06 |
| ORTHOG-SUBSPACE | ✗ | 29.1(±1.99) | 0.53(±0.03) | 34.6(±1.12) | 0.22(±0.01) |
| ORTHOG-SUBSPACE-BN | ✗ | 34.3(±0.31) | 0.35(±0.01) | 31.2(±1.30) | 0.23(±0.02) |
| ORTHOG-SUBSPACE-GN | ✗ | 40.9(±2.31) | 0.11(±0.03) | 38.6(±1.23) | 0.35(±0.02) |
| **LcSP** | ✗ | 24.1(±0.76) | 0.52(±0.01) | 23.0(±0.84) | 0.50(±0.00) |
| **LcSP-GN** | ✗ | 66.7(±0.54) | **0.00**(±**0.00**) | 65.0(±1.14) | **0.00**(±**0.00**) |
| **LcSP-BN** | ✗ | 72.7(±**1.18**) | **0.00**(±**0.00**) | **67.9**(±**0.40**) | **0.00**(±**0.00**) |

angle. The rotation angle is a random value between $[0, \pi]$ and arbitrary selection for different tasks. In this experimental setup, both of the above MNIST datasets contain 20 tasks, each task containing 10,000 samples from 10 classes. Split CIFAR is constructed by splitting CIFAR100 into multiple tasks, where each task contains the data pertaining to five random classes (without replacement) out of the total 100 classes. Split miniImageNet Vinyals et al. (2016) is a subset of ImageNet. In Split miniImageNet, each task contains the data from five random classes (without replacement) out of 100 classes. Both CIFAR100 and miniImageNet contain 20 tasks, each contains 250 samples from each of the five classes.

**Benchmarks for Learning 150 Tasks and 64 Tasks**    This experiment was conducted on two image classification datasets: Permuted MNIST and Permuted CIFAR10. Both Permuted MNIST and Permuted CIFAR10 are obtained by randomly rearranging image pixels. In this experimental setup, Permuted MNIST contains 150 tasks, and Permuted CIFAR10 contains 64 tasks, each containing 10,000 samples from 10 classes.

## 5.2 BASELINES

We compare the proposed method with SOTA based on GOP. As aforementioned, we generalize GOP into COP and SOP. For methods based on the COP strategy, we compare proposed method with OWM Zeng et al. (2019), Adam-NSCL Wang et al. (2021) and GPM Saha et al. (2021). For methods based on the SOP strategy, we compare the proposed method with ORTHOG-SUBSPACE Chaudhry et al. (2020). Moreover, we also compare our method with HAT Serra et al. (2018), EWC Kirkpatrick et al. (2017), ER-Ring Chaudhry et al. (2019), AGEM Chaudhry et al. (2018b) and **K**ernel **C**ontinual **L**earning (KCL) Derakhshani et al. (2021).

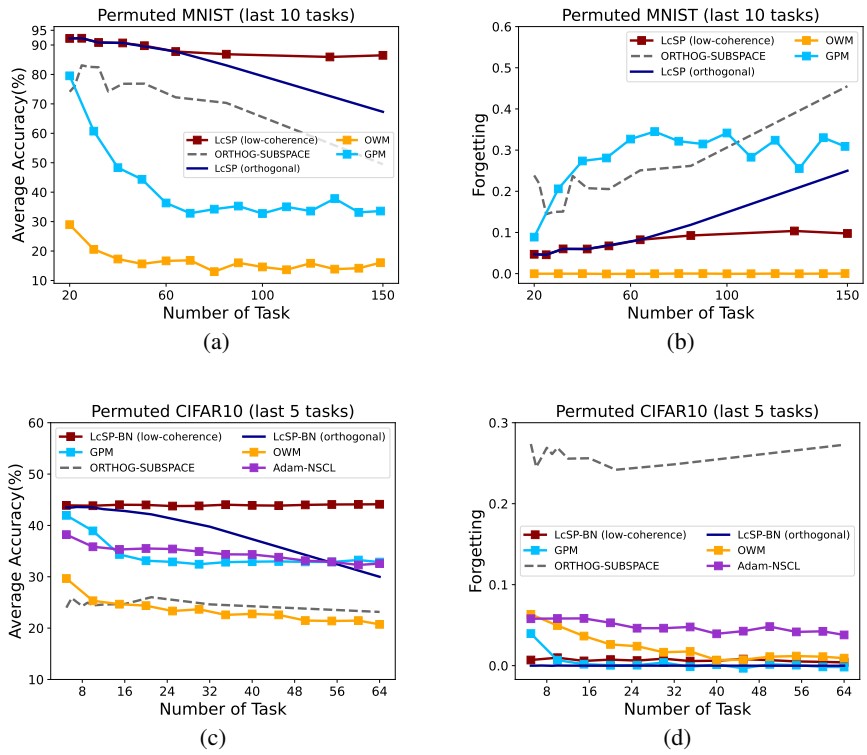

Figure 1: Fig. 1(a) and 1(b) show the average accuracy and forgetting of the last 10 tasks on Permuted MNIST when learning 150 tasks. Fig. 1(c) and 1(d) show the average accuracy and forgetting of the last 5 tasks on Permuted CIFAR10 when learning 64 tasks.

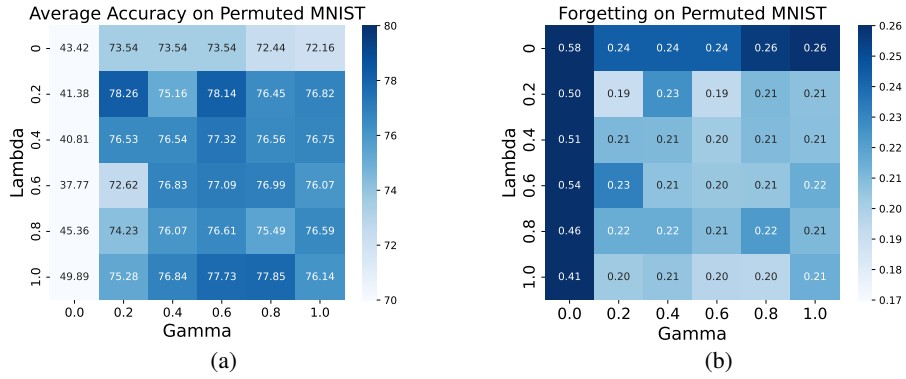

Figure 2: Average accuracy and forgetting for different $\lambda$ and $\gamma$ on Permuted MNIST. A fully connected network with 2 hidden layers, each with 64 neurons, is used for this experiment.

## 5.3 IMPLEMENTATION DETAILS

**Learning 20 Tasks** For experiments on Permuted MNIST and Rotated MNIST, all methods use a fully connected network with two hidden layers, each with 256 neurons, using ReLU activations. For experiments on CIFAR and miniImageNet, all methods use standard ResNet18 architecture except OWM, HAT, and GPM which use AlexNet Krizhevsky et al. (2012). As described in § 4.2, the proposed LcSP makes two changes to the BN layer in standard ResNet18: (1) learning specific BN for each task and (2) replacing BN with GN. We also apply these strategies to ORTHOG-SUBSPACE

for additional comparison. For experiments on MNIST, all tasks share the same classifier. For experiments on CIFAR and miniImageNet, each task requires a task-specific classifier. For all experiments, LcSP does not use episodic memory to store data samples for data replay. For all methods, We use **S**tochastic **G**radient **D**escent (SGD) uniformly. The learning rate is set to $0.01$ for experiments on MNIST and $0.003$ for experiments on CIFAR and ImageNet. Both $\lambda$ and $\gamma$ in Eq. (**??**) are set to $1$. All experiments were run five times with five different random seeds, with the batch size besing $10$.

**Learning** $150$ **Tasks and** $64$ **Tasks** For experiments on Permuted CIFAR10, LcSP uses ResNet18 architecture and applies **the strategy (1)** to change BN in ResNet18. To compare the performance of different GOP strategies, we did not use episodic memory for ORTHOG-SUBSPACE. Except for these changes, other experimental settings are the same as described above.

### 5.4 EXPERIMENTAL RESULTS

**Comparisons of Learning 20 Tasks** Table 1 compares the average accuracy and forgetting results of the proposed **LcSP** and its variants (**LcSP-BN** and **LcSP-GN**) with baselines on the four continual learning benchmarks. Therein, **LcSP-BN** and **LcSP-GN** adopt *strategy* (1) and (2) described in § 4.2, respectively. First, as shown in Table 1, the proposed methods outperform all baselines on MNIST and miniImageNet. On Permuted MNIST and Rotated MNIST, the average accuracy of LcSP surpasses the baselines by $23.8\% \sim 5.6\%$ and $43\% \sim 4.8\%$, respectively. On miniImageNet, the average accuracy of **LcSP-BN** surpasses the baselines by $44.9\% \sim 4.63\%$. On CIFAR100, the proposed **LcSP-BN** achieved a competitive performance with the second highest average accuracy, $3.25\%$ lower than Adam-NSCL, and a forgetting rate of 0. The average accuracy of **LcSP-GN** also outperforms most baselines, being lower than Adam-NSCL, HAT and GPM on CIFAR100 but higher than compared methods on miniImageNet. These results suggest that minimizing the inter-task and intra-task coherence with low-coherence projectors is an effective strategy for solving catastrophic forgetting. Secondly, results on CIFAR100 and miniImageNet also show that BN in ORTHOG-SUBSPACE Chaudhry et al. (2020) and LcSP may change previous tasks' data distribution and lead to catastrophic forgetting. Both strategies (1) and (2) described in § 4.2 can effectively solve this problem.

**Comparisons of Learning 150 Tasks and 64 Tasks** In order to demonstrate the high advantage of proposed methods in learning a long sequence of tasks, the following experiments compare the results with 64 tasks and 150 tasks. Note that, in Fig. 1, LcSP (orthogonal) and LcSP-BN (orthogonal) use orthogonal projectors as comprisons while LcSP (low-coherence) and LcSP-BN (low-coherence) use low-coherence projectors. Fig. 1(a) and 1(b) report the average accuracy and forgetting of last 10 tasks, with learning 150 tasks on Permuted MNIST. Fig. 1(c) and 1(d) report the average accuracy and forgetting of last 5 tasks with learning 64 tasks on Permuted CIFAR10. The average accuracy of all methods, except the proposed LcSP-BN (low-coherence), dramatically degrades or is consistently low as the number of tasks increases. Furthermore, it can be seen from 1(d) that all methods except ORTHOG-SUBSPACE have almost no forgetting. This result indicates that methods using orthogonal projectors gradually lose their learning capacity with increasing number of tasks. The proposed method uses the low-coherence projector to relax the orthogonal restriction, effectively solving this problem. Fig. 2 gives the ablation study and shows the performance of our method with different $\lambda$ and $\gamma$. When $\lambda$ equals $\gamma$, the result of average accuracy on Permuted MNIST reached the highest. Results reach the worst when either $\lambda$ or $\gamma$ equals zero. These results indicate that both inter-task and intra-task coherence should be minimized to solve the plasticity and stability dilemma.

## 6 CONCLUSION

This paper proposed a novel gradient projection approach for continual learning to address gradient orthogonal projection's learning capacity degradation problem. Instead of learning in orthogonal subspace, we propose projecting features and gradients via low-coherence projectors to minimize inter-task and intra-task coherence. Additionally, two strategies have been proposed to mitigate the catastrophic forgetting caused by the BN layer, i.e., replacing BN with GN or learning specific BN for each task. Extensive experiments show that our approach works well in alleviating forgetting and has a significant advantage in maintaining learning capacity, especially in learning long sequence tasks.

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

## A  APPENDIX

In this section, we give the implementation details about the experiments on Permuted MNIST and Permuted CIFAR10, to help readers reproduce these experiments. Additionally, more ablation studies and experimental results are provided here to further support the conclusions and contributions.

### A.1  IMPLEMENTATION DETAILS

The main hyperparameter settings are listed in Table 2 and 3. For the baselines, we adopt the default settings provided in their code to bring out the proper performance. For fair comprision, we use the uniform **Batch size** for all methods.

Table 2: Implementation details for the experiments on Permuted MNIST.

| Method | Optimizer | Momentum | Learning rate | Max epochs | Early stop | Batch size |
|---|---|---|---|---|---|---|
| OWM | SGD | 0.9 | 2.0 | 100 | ✗ | 10 |
| GPM | SGD | 0.9 | 0.01 | 5 | ✗ | 10 |
| ORTHOG-SUBSPACE | SGD | 0.9 | 0.1 | 15 | ✓ | 10 |
| LcSP(low-coherence) | SGD | - | 0.1 | 15 | ✓ | 10 |
| LcSP(orthogonal) | SGD | - | 0.1 | 15 | ✓ | 10 |

Table 3: Implementation details for the experiments on Permuted CIFAR10.

| Method | Optimizer | Momentum | Learning rate | Max epochs | Early stop | Batch size |
|---|---|---|---|---|---|---|
| OWM | SGD | 0.9 | 0.02 | 25 | ✗ | 64 |
| GPM | SGD | 0.9 | 0.01 | 200 | ✓ | 64 |
| ORTHOG-SUBSPACE | SGD | 0.9 | 0.1 | 80 | ✓ | 64 |
| LcSP(low-coherence) | SGD | - | 0.003 | 80 | ✓ | 64 |
| LcSP(orthogonal) | SGD | - | 0.003 | 80 | ✓ | 64 |

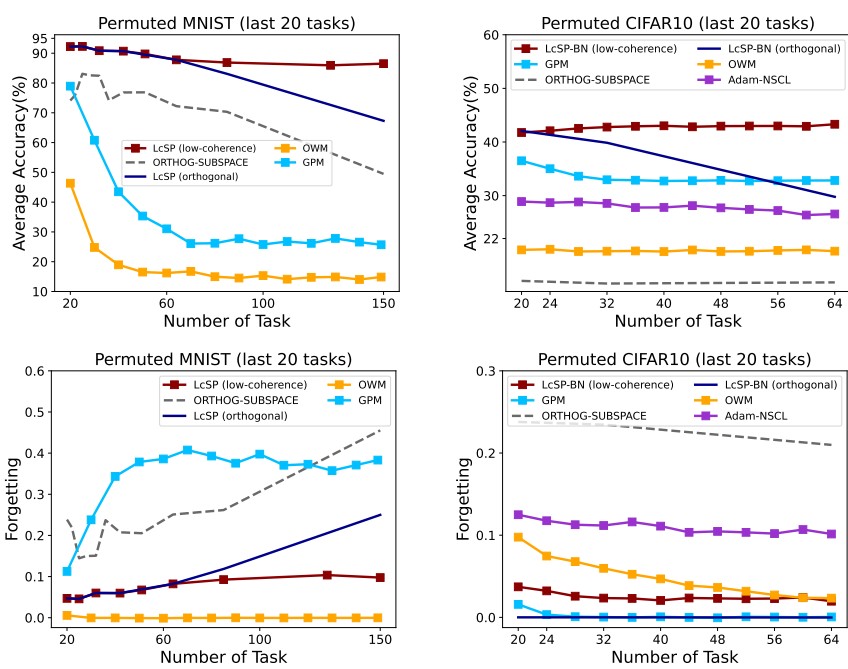

Figure 3: The average accuracy and forgetting of the last 20 tasks on Permuted MNIST (left) and Permuted CIFAR10 (right). The x-axis shows the number of tasks learned and y-axis represents the corresponding average accuracy and forgetting on the last 20 tasks.

## A.2 ABLATION STUDIES AND ADDITIONAL RESULTS

**Additional results on Permuted MNIST and Permuted CIFAR10** Readers may wonder whether our conclusion holds if we evaluate the average performance with more tasks (e.g., the average accuracy and forgetting on the last 20 tasks). As shown in Fig.3, LcSP still outperforms all baselines with a significant advantage. However, the phenomenon of learning capacity degradation in baselines becomes more imperceptible, e.g., the average accuracy of OWM on Permuted CIFAR10 is consistently low, rather than significantly decreasing. To further investigate the learning capacity degradation problem, we give the accuracy of baselines for the last task on Permuted MNIST and Permuted CIFAR10. As shown in Fig.7, all baselines, except LcSP, suffer from this problem with different degrees and show some decrease in accuracy compared to the initial ($66.16\% \sim 24.63\%$ on Permuted MNIST and $24.8\% \sim 3.48\%$ on Permuted CIFAR10). These results suggest that this

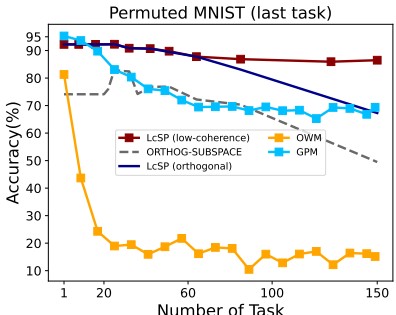 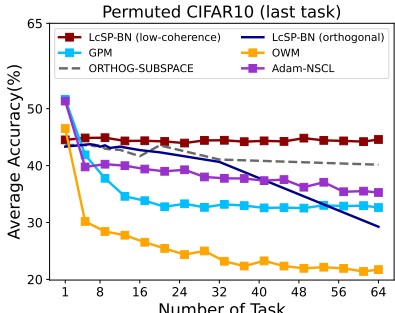

Figure 4: The accuracy of the last task on Permuted MNIST (left) and Permuted CIFAR10 (right), respectively.

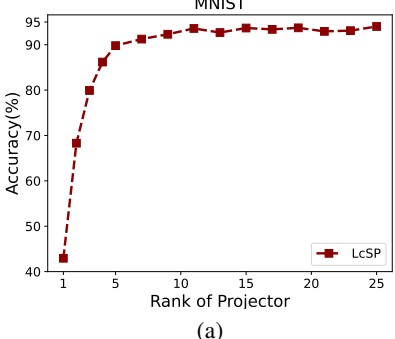 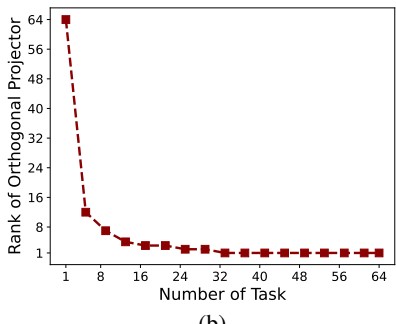

(a)  (b)

Figure 5: Fig.(a) gives the ablation study for different rank of projectors on MNIST. Fig.(b) shows the average rank of all projectors when the number of tasks increasing. Here, the dimension $m$ of features is 64.

problem is the critical factor that results in degrading the performance of GOP-based methods in the case of a large number of tasks.

**Ablation studies and experiments for rank and scale constraints**   Further ablation studies and experiments are conducted to investigate the effects of the rank and scale constraints on the expressive power (plasticity) and stability of DNNs.

The result in Fig.5(a) suggests that projecting features or gradients into subspaces with low dimensions (e.g., lower than 5 in Fig.5(a) ) will decrease the expressive power of DNN. GOP methods and LcSP rely on the projectors to project the features or gradients (or both) into a $d$-dimensional subspace, which can also be considered as a form of dimension reduction. The dimension reduction is motivated by a consensus in the high-dimensional data analysis community, i.e., the data can be summarized in a low-dimensional space embedded in a high-dimensional space, such as a nonlinear manifold Levina & Bickel (2004). The dimension of this low-dimensional space is also known as the intrinsic dimension $D$ Carreira-Perpinán (1997). If the $d$ is too small, e.g., $d \ll D$, important data features will be "collapsed" onto the same dimension. From the perspective of training a DNN, if the gradients are projected into a subspace with a dimension lower than $D$, the DNN cannot activate sufficient parameters to learn the presentation of this task.

Due to the unknown number of tasks to be learned, the limited dimensions of the features, and the strict orthogonality constraint between projectors, the orthogonal projector cannot be constrained to a fixed-rank manifold. As shown in Fig.5(b), the rank of the orthogonal projector decreases as the number of tasks increases. Therefore, methods using orthogonal projectors usually ignore the intrinsic dimension of data (features or gradients) and finally lead to suffering from the learning capacity degradation problem. In contrast to GOP methods, LcSP relaxes the orthogonality constraint

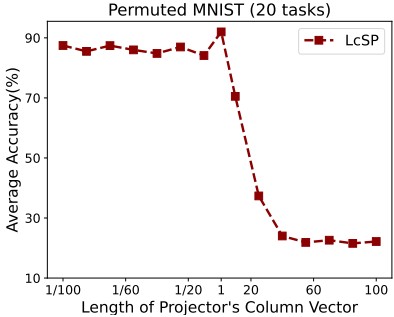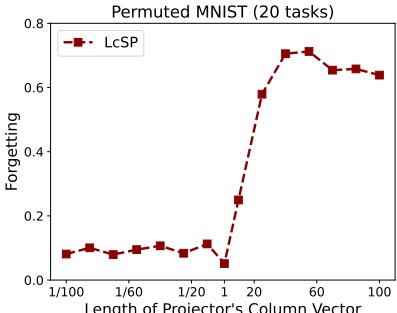

Figure 6: The average accuracy (left) and forgetting (right) of the last 20 tasks with different scales of columns on Permuted MNIST.

and meets the rank constraint by optimizing intra-task coherence on the Oblique manifold and thus does not suffer from this problem.

Finally, Fig.6 gives an ablation study for scale constraints on the projector's columns. In Fig.6, when the columns of the projector have unit length, the average accuracy reaches the highest. Result gets worse when the length of the projector's columns is too small or too large.

Table 4: Efficiency analysis.

|  | OWM | GPM | Adam-NSCL | ORTHOG-SUBSPACE | LcSP |
|---|---|---|---|---|---|
| FLOPs | 26,676,256 | 23,202,880 | 558,538,752 | 556,708,864 | 528,965,376 |
| Time(s) | 47.614 | 5.174 | 47.685 | 87.623 | 25.496 |
| Epochs | 25 | 200 | 80 | 80 | 80 |
| Mean_inference_time(ms) | 1.249 | 1.152 | 10.089 | 3.957 | 5.893 |

Based on the RTX3060 12G, we tested the efficiency of LcSP and compared methods on Permuted CIFAR10 (which contains 64 tasks, and each image's size is $3 \times 32 \times 32$), using AlexNet for OWM and GPM, and resnet18 for the other methods. We provide 4 metrics to assess the efficiency of the method: floating point operations **FLOPs**, time spent per epoch during training **Time (s)**, the number of epochs needed to train to convergence **Epochs**, and the average time spent on inference **Mean_inference_time (ms)**. The detailed results are listed in Table.4.

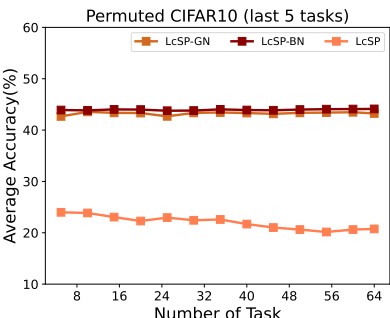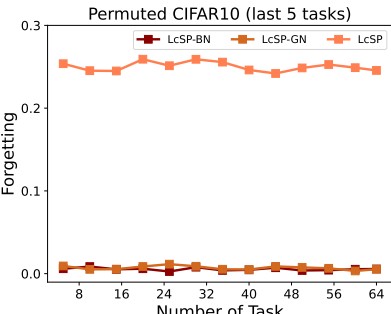

Figure 7: The average accuracy (left) and forgetting (right) of the last 5 tasks on Permuted CIFAR10. All methods use the low-coherence projector.

### A.3 METHOD ANALYSIS

**The mechanism of low-coherence learning and the difference with orthogonal learning** Recall the Lemma 1 described in Section 4.3. **Lemma 1.** *Assume that $f$ is fed the data of task $\mathcal{T}_t$ ($q < t$), then $f$ can effectively overcomes catastrophic forgetting if*

$$z_{q,q}^l \approx z_{q,t}^l, \quad \forall q \leq t$$

. Here, $z_{q,t}^l$ denotes the output feature of $l$-th layer when data from task $\mathcal{T}_q$ are fed into the DNN that has been trained on task $\mathcal{T}_t$. Let us consider the case where there are only two tasks, i.e., $q = t - 1$. By appling the LcSP's projection strategy, we have

$$z_{q,t}^l = z_{q,q}^l + x_{q,t}^l \Delta W_t^l P_q^l = z_{q,q}^l + x_{q,t}^l g_t^l P_t^l P_q^l$$

. Here, $g_t^l$, $x_{q,t}^l$ and $P_t^l$ denote the gradient, input data, and projection matrix (which is a symmetric matrix) of task $\mathcal{T}_t$ in $l$-th layer, respectively. The term $x_{q,t}^l g_t^l P_t^l P_q^l$ can be thought of as the forgetting, which changes the output of DNN to previous data. Since we cannot change $x_{q,t}^l$ and $g_t^l$, the key to reduce $x_{q,t}^l g_t^l P_t^l P_q^l$ is to minimize $P_t^l P_q^l$. If $P_t^l$ is orthogonal to $P_q^l$, the forgetting is equal to 0. Let us consider reducing forgetting when $P_t^l$ cannot be orthogonal to $P_q^l$ (e.g., the number of tasks is large and the dimensionality of the space is limited). If we optimize $P_t^l P_q^l$ directly, we may get $P_t^l = 0$, or the column scale of $P_t^l$ is too small. To find a useful $P_t^l$, LcSP constrains the length of the column vector to be 1, i.e., to find $P_t^l$ on the Oblique manifold ($\mathcal{OM}$). To reduce forgetting, we optimize the inter-task coherence $\mu(P_t^l, P_q^l)$ on $\mathcal{OM}$ so that the maximum entry of $P_t^l P_q^l$ is minimized.

In the worst case, we may obtain $P_t^l$ consisting of only one column vector (i.e., $P_t^l$ with rank 1). To avoid this situation, we optimize the intra-task coherence $\mu(P_t^l)$ along with inter-task coherence, forcing the projector to satisfy the rank constraint in order to maintain the learning capacity of DNN (as shown in Fig. 5(a), the rank of projector affects the plasticity of DNN).

In general, low-coherence projection, which can be seen as a relaxed orthogonal constraint, aims to balance plasticity and stability better, motivated by our observation that orthogonal projections reduce the plasticity of the model and make the DNN unable to adapt to new environments.

