# OpenReview forum: "Continual Learning In Low-coherence Subspace: A Strategy To Mitigate Learning Capacity Degradation"
_ICLR.cc/2023/Conference — Submitted to ICLR 2023_

### Official Review · Reviewer_8QtN · 2022-10-19

**Confidence:** 2
**Correctness:** 3
**Technical Novelty And Significance:** 2
**Empirical Novelty And Significance:** 2
**Recommendation:** 5

**Clarity, Quality, Novelty And Reproducibility:**

Writing clarity can be improved. For example, how to set the O_1 for the first task? The algorithm 1 should be revised in a more clear way. For example, line 1 in Alg. 1 is not necessary. The settings of \alpha and \beta in algorithm 1 should be clear.

**Strength And Weaknesses:**

Strength:

(1) The proposed approach tackles the challenging continual learning task, and proposed a gradient projection method, based on low-coherence, instead of orthogonality.

(2) The experiments on Permuted MNIST, Rotated MNIST, Split CIFAR100 and Split miniImageNet show the improved results compared with some baseline methods, e.g., EWC, GPM, OWM, etc.

Weakness:

(1)  The proposed method is based on construction of the projection matrix by minimizing the coherence of the projection matrix with respect to the projection matrices of previous task, and also within the current task projection matrix.  Based on the sect. 4.1, it is unclear how to generate the projection matrix for the first task. Moreover, for the projection matrix formulation in eq. (6), it seems that the current O_t is purely determined based on the previous task O_i (i = 1, ..., t-1). It seems that the task data (or their features) are not utilized in the determination of O_t. How to guarantee that that learned O_t is optimal without seeing the task data or their features?

(2) According to the results in Table I, the LcSP achieves significantly lower results than the compared methods, on Split CIFAR100, Split miniImageNet. But LcSP-GN and LcSP-BN are better than the compared methods. How to explain this results? It seems that the learning BN for each task is essential to the good results on these two tasks. Are these compared methods also use the normalization technique as the LcSP-GN and LcSP-BN? Moreover, the comparisons of LcSP, LcSP-GN and LcSP-BN should be also included for other datasets.

(3) The comparisons are insufficient, e.g., the Adam-NSCL is also an typical continual learning method based on orthogonal projection, which should be also compared. The compared methods in Table 1 for different datasets are not consistent, for example, GPM and OWM are not compared for the two versions of MNIST dataset.

**Summary Of The Paper:**

This paper proposed a continual learning method by constructing low-coherence subspace projector for each new task. Given a new task, the projector matrix is constructed by minimizing the coherence to the previous task projectors and within the current projector to be optimized, in the oblique manifold. The proposed approach is evaluated for task incremental learning tasks, especially when the number of tasks is large (e.g., 64, 150). The results show improvements over the baseline methods, including, EWC, OWM, GPM, etc.

**Summary Of The Review:**

My major concerns on this work are some unclear details, e.g., the question (1) in weakness, and insufficient comparisons in experiments. Moreover, there are some unclear points in the experiment results, especially on the effect of normalization techniques in the results.

---

> ### Author Response · Authors · 2022-11-18
> **Respone to Reviewer 8QtN (Question 1)**
>
> **Question 1. The proposed method is based on construction of the projection matrix by minimizing the coherence of the projection matrix with respect to the projection matrices of previous task, and also within the current task projection matrix. Based on the sect. 4.1, it is unclear how to generate the projection matrix for the first task. Moreover, for the projection matrix formulation in eq. (6), it seems that the current $O_t$ is purely determined based on the previous task $O_i$ (i = 1, ..., t-1). It seems that the task data (or their features) are not utilized in the determination of $O_t$. How to guarantee that that learned $O_t$ is optimal without seeing the task data or their features?**
>
> **Answer to Question 1.** We highly appreciate this comment. We have modified **the Eq. 6 on page 4 and Algorithm 1 on page 5** in the revision to clarify how to generate the projection matrix for the first task. The corresponding revision has already been uploaded to the OpenReview system, which is shown as follows.
>
> Specifically, we consider this confusion caused by the lack of boundary condition. Taking the boundary condition into consideration, we rewrite Eq. (6) (the cost function in Alg.1) as follows:
>
> \begin{aligned}\mathbf{J}(O_t) &= \begin{cases}
> \lambda \cdot \sum_ {i = 1}^ {t-1} \mu(O_i O_i^\top,O_t O_t^\top) + \gamma \cdot \mu(O_t O_t^\top,O_t O_t^\top), & t>1 \\\\
> \mu(O_t O_t^\top,O_t O_t^\top), & t=1
> \end{cases}\\\
> O_t &= \arg \min \mathbf{J}(O_t), \quad \text {s.t. } \quad {O_t} \in \mathcal{OB}(m,d_t).\end{aligned}
>
> **This cost function can still be optimized by following Alg.1. What will we obtain for the first task?**  When $t = 1$, we don't need to consider the inter-task coherence since there is no knowledge interference. By optimizing intra-task coherence, we will finally obtain an orthogonal projector $P_t = O_tO_t^\top$, $P_tP_t^\top = I_k$, where $k$ represents the dimension of the projector.
>
> **How to guarantee that that learned $O_t$ is optimal without seeing the task data or their features?**   According to Eq.(6) and Alg.1, we can construct $O_t$ without seeing any data (even before learning any task) as long as we construct $O_1,...,O_{t-1}$ in sequence.

---

> > ### Author Response · Authors · 2022-11-18
> > **Respone to Reviewer 8QtN (Question 2)**
> >
> > **Q.2. According to the results in Table I, the LcSP achieves significantly lower results than the compared methods, on Split CIFAR100, Split miniImageNet. But LcSP-GN and LcSP-BN are better than the compared methods. How to explain this results? It seems that the learning BN for each task is essential to the good results on these two tasks. Are these compared methods also use the normalization technique as the LcSP-GN and LcSP-BN? Moreover, the comparisons of LcSP, LcSP-GN and LcSP-BN should be also included for other datasets.**
> >
> > **A.2.** Thank you for this valuable comment. In the following, we give more explanations for these results and our motivation.
> >
> > **The motivation to use these normalization techniques:** In our research, we observe that the proposed LcSP **works well when using MLP (without BN layer)** but **suffers from catastrophic forgetting when using a BN-based model** (e.g., resnet18). In order to retain the advantages of the BN layer (e.g., as described in **Section 4.2**, the BN layer can accelerate convergence and make training more stable), we explored the reason and proposed solutions for this problem, i.e., (1) learning task-specific BN or (2) replacing BN with GN, and conduct experiments on Split CIFAR100 and Split miniImageNet.
> >
> >
> > As shown in Table 1, the LcSP suffered catastrophic forgetting (0.52 forgetting) when using the BN-based model, whereas both strategies effectively addressed this problem (0 forgetting).
> >
> > **The reason for BN causing forgetting:** Here, we discuss the reason for this problem in more detail. Let $x$ denote input data, $\mu$ denote the mean of seen data, and $\sigma$ denote the variance of seen data. **In the process of forward inference**, BN normalizes $x$ and applies an affine mapping, which can be described as follows:
> >
> > $\displaystyle \begin{array}{l} \widehat{x} \leftarrow \frac{x - \mu}{\sqrt{\sigma^{2} + \epsilon}}, \\\ y_{i} \leftarrow \gamma \widehat{x} + \beta \equiv \operatorname{BN}_ {\gamma, \beta} \left(x_ {i} \right) \end{array}.$
> >
> > **Here, as global mean and variance, $\mu$ and $\beta$ will be updated when new data arrive, changing the normalization of previous data. This change interferes with the knowledge that LcSP has learned and causes catastrophic forgetting.** To address this problem in continual learning, we should calculate and save $\mu$ and $\beta$ for the individual task rather than all tasks (e.g., task-specific BN) or use the local rather than global means and variances (e.g., GN, which can work well **when batch size is small**).
> >
> > This normalization technique is part of LcSP, and we have not applied it to other compared methods. To the best of our knowledge, some GOP methods use different ways to handle the BN layer, e.g., GPM use BN with no track running states, and Adam-NSCL uses EWC to regularize the parameter of the BN layer.
> >
> > **Moreover, the comparisons of LcSP, LcSP-GN and LcSP-BN should be also included for other datasets:**  As you suggested, we include LcSP and LcSP-GN for Permuted CIFAR10 in the revision. The additional experimental results are shown **in Fig.7 on page 15**. **The corresponding revision has already been uploaded to the Openreview system**. **Please note that** for experiments on MNIST, we do not apply this normalization technique as we do not use BN-based model.

---

> > ### Author Response · Authors · 2022-11-18
> > **Respone to Reviewer 8QtN (Question 3)**
> >
> > **Q.3. The comparisons are insufficient, e.g., the Adam-NSCL is also a typical continual learning method based on orthogonal projection, which should be also compared. The compared methods in Table 1 for different datasets are not consistent, for example, GPM and OWM are not compared for the two versions of MNIST dataset.**
> >
> > **A.3.** Thank you for this comment. As you suggested, we have added the additional experimental results in the revision. **Please see Table.1 and Fig.1 on pages 7 and 8 in the uploaded revision.**
> >
> > Specifically, we compared Adam-NSCL on Permuted MNIST, Permuted CIFAR10, Split CIFAR100 and Split miniImageNet. In addition, we also compared OWM and GPM on MNIST to keep the comparison across datasets as consistent as possible.
> >
> > The following are some additional experimental results from the revision. **Please note that all tasks share the same classifier for experiments on MNIST.**
> >
> > **Table.1: Results on Permuted MNIST.**
> >
> > |  Method   | average accuracy (%) | forgetting |
> > | :-------: | :------------------: | :--------: |
> > |    GPM    |        80.54         |    0.16    |
> > |    OWM    |        46.35         |    0.01    |
> > | Adam-NSCL |        26.44         |    0.70    |
> > |   LcSP    |         92.2         |    0.05    |
> >
> > **Table.2: Results on Rotated MNIST.**
> >
> > |  Method   | average accuracy (%) | forgetting |
> > | :-------: | :------------------: | :--------: |
> > |    GPM    |        80.77         |    0.16    |
> > |    OWM    |        44.78         |    0.01    |
> > | Adam-NSCL |        42.73         |    0.56    |
> > |   LcSP    |         86.6         |    0.08    |
> >
> > **Table.3: Results on Split CIFAR100.**
> >
> > |  Method   | average accuracy (%) | forgetting |
> > | :-------: | :------------------: | :--------: |
> > |    OWM    |        50.94         |    0.30    |
> > |    GPM    |        72.48         |    0.00    |
> > | Adam-NSCL |        75.95         |    0.04    |
> > |  LcSP-BN  |        72.70         |    0.00    |
> >
> >
> >
> > **Table.4: Results on Split MiniImageNet.**
> >
> > |  Method   | average accuracy (%) | forgetting |
> > | :-------: | :------------------: | :--------: |
> > |    OWM    |          -           |     -      |
> > |    GPM    |        60.41         |    0.00    |
> > | Adam-NSCL |        63.27         |    0.06    |
> > |  LcSP-BN  |        67.90         |    0.00    |

---

### Official Review · Reviewer_em94 · 2022-10-23

**Confidence:** 2
**Correctness:** 3
**Technical Novelty And Significance:** 2
**Empirical Novelty And Significance:** 2
**Recommendation:** 6

**Clarity, Quality, Novelty And Reproducibility:**

Low-coherence subspaces projection can be a good alternative to GOP. It is novel.
More thorough analysis can be provided for the severity of GOP's learning capacity degradation issue. The authors' claim "their learning capacity is gradually degraded as the number of tasks increases and eventually becomes unlearnable" needs solid analysis and evidence. Likewise analysis and evidence for the proposed method does not suffer from this should be provided too.

GOP should be used when the network has sufficient learning capacity as Chaudhry et al. (2020) pointed out that “We assume that the network is sufficiently parameterized, which often is the case with deep networks, so that all the tasks can be learned in independent subspaces.” If learning capacity can not catch up with the demands of increasing tasks, perhaps we can try to increase capacity on the fly? One way is to grow the network's depth adaptively [1*].

[1*] Online Deep Learning: Learning Deep Neural Networks on the Fly, by Doyen Sahoo, Quang Pham, Jing Lu, Steven C. H. Hoi, IJCAI 2018.


**Details Of Ethics Concerns:**

No ethic concerns.

**Strength And Weaknesses:**

Strength:
1) learning in low-coherence subspaces can be an alternative to GOP.
2) the experimental results are impressive.

Weakness:
1) The authors believe that GOP causes learning capacity degradation and its learning capacity "is gradually degraded as the number of tasks increases and eventually becomes unlearnable". This should be more thoroughly analysed and supported by ablation study (curves on 64 tasks and 150 tasks with other methods are not sufficient to serve this purpose).
2) The proposed method's forgetting performance on different datasets and settings vary widely from 0 forgetting to the state of being beaten by one or two competitors (e.g. see Fig. 1 (b) (d)).
3) Coherence metric can be restrictive. Babel function which extends cross-correlation between pairs of columns to the cross-correlation from one column to a set of other columns might be a better alternative.


**Summary Of The Paper:**

This paper proposes to learn new tasks in low-coherence subspaces rather than orthogonal subspaces to mitigate catastrophic forgetting in continual learning. The authors believe that Gradient Orthogonal Projection (GOP) (though helps battle catastrophic forgetting) causes  learning capacity degradation and its learning capacity "is gradually degraded as the number of tasks increases and eventually becomes unlearnable".

To learn new tasks in low-coherence subspaces, the authors propose a unified cost function to seek projectors and develop a gradient descent algorithm (on the Oblique manifold) by jointly minimizing inter-task coherence and intra-task coherence. The original mutual coherence (often used in compressive sensing) of a matrix is the maximum absolute value of the cross-correlations between the matrix's columns. The authors extend the concept of coherence to two matrices to capture the maximum absolute value of the cross-correlations for the columns from two matrices. To deal with catastrophic forgetting heightened by batch normalisation (BN), the authors use two strategies: (1) learning specific BN for each task, or (2) using group normalisation (GN) instead of BN.

The authors did experiments on Permuted MNIST, Rotated MNIST, Split CIFAR100 and Split miniImageNet. For 20 tasks (a typically setting), the proposed method achieved best accuracies on Permuted MNIST and Rotated MNIST, and 3nd best forgetting on Permuted MNIST and 2rd best forgetting on Rotated MNIST. The proposed method was reported to achieve best accuracies and zero forgetting (surprising to see zero forgetting) on Split CIFAR100 and Split miniImageNet.

To show other methods would struggle with increasing tasks, the authors did experiments on 64 tasks and 150 tasks where the proposed method shows a clear advantage on accuracies, but not necessarily on forgetting (the proposed method is reported to achieve  best accuracies on both Permuted MNIST and Permuted CIFAR10, and 2nd best forgetting on Permuted MNIST and 3rd best forgetting on Permuted CIFAR10).

**Summary Of The Review:**

At this stage, I am happy for this paper to be accepted.

---

> ### Author Response · Authors · 2022-11-18
> **Response to Reviewer em94 (Question 1)**
>
> **Question 1. The authors believe that GOP causes learning capacity degradation and its learning capacity "is gradually degraded as the number of tasks increases and eventually becomes unlearnable". This should be more thoroughly analysed and supported by ablation study (curves on 64 tasks and 150 tasks with other methods are not sufficient to serve this purpose).**
>
> **Answer to Question 1.** We highly appreciate this constructive suggestion. We agree that this paper should provide more ablation study and analysis to support the claim **"is gradually degraded as the number of tasks increases and eventually becomes unlearnable"**, and we have done this in the revised Appendix which can be found in the system.
>
> The Appendix provides an additional ablation study to investigate the learning capacity degradation problem. Specifically, we first investigated the relationship between model plasticity and the number of tasks. As shown in **Fig.4 on page 14**, which reports the accuracy of compared methods on new tasks, the performance of compared methods gradually decreases as the number of tasks increases. In addition, we investigated the effect of projector rank on model plasticity. As shown in **Figure 5(a) on page 14**, the plasticity of the model increases with increasing projector rank. Finally, **Figure 5(b)** depicts the relationship between projector rank and the number of tasks under strict orthogonality conditions, showing that projector rank decreases with increasing tasks.
>
> On **page 14**, the phenomenon in **Fig.5 (a)** is explained from the perspective of the data and parameterization as follows. Let $d$ denote the rank of the projector and $D$ denote the intrinsic dimension of data. **" If the d is too small, e.g., d *≪* *D*, important data features will be ”collapsed” onto the same dimension. From the perspective of training a DNN, if the gradients are projected into a subspace with a dimension lower than *D*, the DNN cannot activate sufficient parameters to learn the presentation of this task."**
>
> **In the following on page 14**, we intuitively explain the phenomenon in **Fig.5 (b)**: **"Due to the unknown number of tasks to be learned, the limited dimensions of the features, and the strict orthogonality constraint between projectors, the orthogonal projector cannot be constrained to a fixed-rank manifold."** Therefore, the average rank of all projectors inevitably becomes lower and lower.
>
> **Conclusion:** Combining these observations and results, we argue that preserving the rank of the projector is the key to balancing plasticity and stability. Inspired by this, we propose the LcSP, which uses rank-preserving, low-coherence projectors. Extensive experimental results show that LcSP balances plasticity and stability more effectively in learning long task sequences.

---

> > ### Author Response · Authors · 2022-11-18
> > **Response to Reviewer em94 (Question 2-3)**
> >
> > **Question 2. The proposed method's forgetting performance on different datasets and settings vary widely from 0 forgetting to the state of being beaten by one or two competitors (e.g. see Fig. 1 (b) (d)).**
> >
> > **Answer to Question 2.** Thank you for this valuable comment. In the following, we give more explanation about this variation.
> >
> > (1) **Whether LcSP can achieve 0 forgetting depends on the number of tasks and model capacity.** By choosing the proper projector rank, LcSP can achieve 0 forgetting with a small number of tasks (e.g., as shown in **Table.1 on page 7**, LcSP can learn without forgetting on 20 tasks).
> >
> > (2) **When the number of tasks is large, LcSP pursues a better balance of plasticity and stability, ensuring the learning capacity for new tasks while inevitably suffering from forgetting.** In this case, LcSP preserves the rank of the projector to keep the learning capacity, and **preserving the rank of the projector makes projectors lose their orthogonality and obtain a low coherence instead.** Although one or two competitors beats its forgetting performance, LcSP performs far better than other methods in accuracy. In this case, the forgetting performance of LcSP using the low-coherence projector (corresponding to the **LcSP (low-coherence)** in **Fig.1**, depicted by the dark red line) is consistently low or with little variation as the task increases.
> >
> >
> >
> >
> > **Answer to Question 3.** We greatly appreciate this valuable suggestion. Thank you for bringing the Babel function to our attention. We have modified **the Eq.6 on page 6** in a more restrictive way. **The revision has been uploaded, and you can find them in this system.**
> >
> > Here are our modifications to Eq.6 on page 6.
> >
> > Specifically, the Babel function [1], measuring the maximal total coherence between an atom and a collection of other atoms, can be described as follows.
> >
> > $\displaystyle B(p)=\max_ {\Lambda,|\Lambda|=p} \max_ {i \notin \Lambda} \sum_ {j \in \Lambda} \frac{\left | \left \langle \mathbf{d}_ {i}, \mathbf{d}_ {j} \right \rangle \right |}{\left \| \mathbf{d}_ {i} \right \| \left \| \mathbf{d}_ {j} \right \|}$.
> >
> > By combining the coherence metric $ \displaystyle \mu(M) = \max_ {j < k} \frac{ | \langle M_j, M_k \rangle |} {\| M_j \|_ {2} \| M_k \|_ {2}}$ and Babel function, we can simplify Eq. 6 to the following form:
> >
> > $\mathbf{J}(O_t) = B(O_tO_t^\top) + \mu(O_tO_t^\top)$.
> >
> > [1] Huan Li and Zhouchen Lin. Construction of incoherent dictionaries via direct babel function
> > minimization. In Asian Conference on Machine Learning, pp. 598–613. PMLR, 2018.

---

### Official Review · Reviewer_FMJV · 2022-10-25

**Confidence:** 5
**Correctness:** 4
**Technical Novelty And Significance:** 2
**Empirical Novelty And Significance:** Not applicable
**Recommendation:** 5

**Clarity, Quality, Novelty And Reproducibility:**

The whole paper is generally written clearly, and the work is with good reproducibility. The details on the task-specific BN can be improved for better reproducibility. The analyses, discussion, and introduction for the settings can be improved. The novelty and contribution are not significant enough.

**Strength And Weaknesses:**

Strength
- This paper proposes to learn the projections in the low-coherence subspace, instead of the orthogonal subspace, which lets the model capacity be used better, while reducing the parameter interference, with less capacity degradation. This is well motivated.

- The experiments prove the claim about the benefits of learning in the low-coherence subspace, compared to learning with the orthogonal subspace.

- The paper is generally written clearly, with some points that can be improved, as discussed in the following.

Weakness
- The work can be seen as an extension of the orthogonal subspace based continual learning paper (Chaudhry et al. 2020). The main difference is the projectors are learned on the Oblique manifold instead of the Riemannian manifold, which enables learning in the low-coherence subspace with some overlaps between the tasks. It brings limited novelty for continual learning.
- Intuitively, the low-coherence subspace learning can be seen as a “relaxation” of the orthogonal subspace learning. The mechanism of low-coherence learning and the difference with orthogonal learning is not clearly analyzed and discussed.
-- Specifically, some questions can be asked and discussed. For example, comparing with the orthogonal subspace learning, how the method alleviates the forgetting issue while not totally isolating the subspaces.
- The proposed method is limited to only the task-incremental setting where the task identifiers are required in both training and testing phases, as shown on page 3 and experiments. It further influences the significance of the work, considering the insignificant technical novelty, although this is with the same setting as (Chaudhry et al. 2020). The authors may further
- Some detailed questions:
-- What is the gap between the implementation and the analysis on page 6?
-- How is the efficiency of the proposed method? How is the running time compared to other methods?


**Summary Of The Paper:**

Regularizing the subspace of the network representation has been used in continual learning to alleviate forgetting. To mitigate learning capacity degradation caused by orthogonal projection/regularization, this paper proposes to learn in low-coherence subspace. It learns task-specific projectors to represent each task in subspace with low coherence to others. To learn in low-coherence subspace, the projectors are learned on the Oblique manifold. The proposed method performs better than the compared methods (especially the orthogonal subspace learning method) on the task-incremental setting (with known task identifiers).

**Summary Of The Review:**

The whole paper is with a clear motivation and representation overall. The novelty and the significance of the contribution can be improved. Although the experiments can prove the method can work well and better than the orthogonal subspace learning method, the experiments are limited to a very specific and even simple setting in CL, which influences the generality and significance.

After checking all the reviews and responses, I would like to keep my original score - below the acceptance line.
The concept of using low-coherence subspace projection in continual learning is interesting. However, the contribution is relatively limited, considering the relationship with the methods working on orthogonal subspace and the restricted task-incremental setting (i.e., the requirement on task id). The technical details and explanations of the experimental results (in Table 1) are unclear and confusing, as indicated by 8QtN. It is blurry what is the main role of the low-coherence projection on performance improvement, compared with the normalization trick.

---

> ### Author Response · Authors · 2022-11-18
> **Response to Reviewer FMJV (Question 1)**
>
> **Question 1.** **The work can be seen as an extension of the orthogonal subspace based continual learning paper (Chaudhry et al. 2020). The main difference is the projectors are learned on the Oblique manifold instead of the Riemannian manifold, which enables learning in the low-coherence subspace with some overlaps between the tasks. It brings limited novelty for continual learning.**
>
> **Answer to Question 1.** Thank you for your valuable comment. We are here to elaborate more and make the novelty more clear from both the research questions and the methodology perspective.
>
> First, we explain the novelty of the work from the perspective of the research problem. The main claim of continual learning is to find an algorithm that allows a DNN to learn continuously over an infinite stream of data, i.e., a common assumption is that the number of tasks is unknown or infinite. Thus, the learning capacity of tasks, i.e., the number of tasks that a model can learn without forgetting, is the key to continual learning methods. However, in practical scenarios, continuously learning  a long sequence of tasks challenges most continual learning methods, i.e.,
> achieving the balance of plasticity and stability in an infinite data stream is a challenge for most works, especially the regularization-based methods, including various orthogonal projection-based methods, shown in **Section 2** in the submitted manuscript.
>
> Orthogonal projection is a popular and effective way to reduce the interference of new knowledge from old knowledge and many methods have been already developed, as shown in Section 2 in the submitted manuscript. However, **this kind of method suffers from the learning capacity degradation problem.** Extensive experimental results show that orthogonal projection based methods are unsatisfactory **in learning long task sequences**, e.g., their performance in average accuracy dramatically decreases as the task increases (e.g., when the number of tasks exceeds 30 on Permuted MNIST). Here, this degradation in average accuracy is caused by the orthogonal projection rather than catastrophic forgetting.
>
> This problem is an almost **unexplored issue**. To the best of our knowledge, this is the first work to specifically discuss the issue and give a solution from both the algorithm and the theoretical analysis.
>
> Secondly, from the **methodology perspective**, LcSP differs from ORTHOG-SUBSPACE (Chaudhry et al. 2020) in three respects.
>
> **(1)** In order to solve the learning capacity problem of orthogonal projection based methods, the proposed LcSP uses the low-coherence optimizing projector on the Oblique manifold rather than the orthogonal projector. By optimizing inter-task and intra-task coherence while preserving projector rank on the oblique manifold, the LcSP finds a better transfer-interference trade-off, consolidating knowledge while maintaining the capacity to adapt to new environments. We proposed **a novel algorithmic solution (LcSP) associated with its geometric optimization (optimization on Oblique manifold) and the theoretical analysis**. Experiments on Permuted CIFAR10 (64 tasks) and Permuted MNIST (150 tasks) show that LcSP performs much better in average accuracy when learning a long task sequence.
>
> **(2)** Compared to ORTHOG-SUBSPACE, projecting only the feature of the last layer, the proposed LcSP uses a layer-wise projection strategy for every layer, which means that every layer's output feature will be projected. Experimentally, this projection strategy has been verified to be more effective in reducing catastrophic forgetting, performing better than ORTHOG-SUBSPACE.
>
> **(3)** The BN layer problem is not addressed in [1], while this work proposed two techniques to overcome the forgetting caused by the BN layer. Experiments on Split CIFAR100 and Split miniImageNet fully demonstrate the effectiveness of these methods. Furthermore, the results in Table.1 show that these techniques can also improve the performance of ORTHOG-SUBSPACE.
>
> [1] Arslan Chaudhry, Naeemullah Khan, Puneet Dokania, and Philip Torr. Continual learning in low-rank orthogonal subspaces. *Advances in Neural Information Processing Systems*, 33:9900–9911, 2020.

---

> > ### Author Response · Authors · 2022-11-18
> > **Response to Reviewer FMJV (Question 2)**
> >
> > **Question 2. Intuitively, the low-coherence subspace learning can be seen as a “relaxation” of the orthogonal subspace learning. The mechanism of low-coherence learning and the difference with orthogonal learning is not clearly analyzed and discussed. -- Specifically, some questions can be asked and discussed. For example, comparing with the orthogonal subspace learning, how the method alleviates the forgetting issue while not totally isolating the subspaces.**
> >
> > **Answer to Question 2.** Thank you for this valuable comment. As you suggested, in the following, we give more explanations for the mechanism and difference between orthogonal subspace learning and low-coherence subspace learning. **(We have added these explanations in the Appendix A.3 on page 15-16. The corresponding revision has already been uploaded to the OpenReview system)**
> >
> > Recall the Lemma 1 described in **Section 4.3.**
> >
> > **Lemma 1.** *Assume that $f$ is fed the data of task $\mathcal{T}_t$ $(q < t)$, then $f$ can effectively overcomes catastrophic forgetting if*
> >
> > $z_{q, q}^{l} \approx z_{q, t}^{l}, \quad \forall q \leq t$.
> >
> > Here, $z^l_{q,t} $ denotes the output feature of $l$-th layer when data from task $\mathcal{T}_ {q}$ are fed into the DNN that has been trained on task $\mathcal{T}_{t}$. Let us consider the case where there are only two tasks, i.e., $q = t - 1$. By appling the LcSP's projection strategy, we have
> >
> > $z^l_{q,t} = z^l_{q,q} + x^l_{q,t}\Delta W_t^lP^l_q =z^l_{q,q} + x^l_{q,t}g^l_tP^l_tP^l_q$.
> >
> > Here, $g^l_t$, $x^l_{q,t} $ and $P^l_t$ denote the gradient, input data, and projection matrix (which is a symmetric matrix) of task $\mathcal{T}_ {t}$ in $l$-th layer, respectively. The term $x^l_{q,t} g^l_{t} P^l_{t} P^l_{q}$ can be thought of as the forgetting, which changes the output of DNN to previous data. Since we cannot change $x^l_{q,t}$ and $g^l_{t}$, the key to reduce $x^l_{q,t}g^l_tP^l_tP^l_{q}$ is to minimize $P^l_tP^l_q$.
> >
> > If $P^l_t$ is orthogonal to $P^l_q$, the forgetting is equal to 0. Let us consider reducing forgetting **when $P^l_t$ cannot be orthogonal to $P^l_q $** (e.g., the number of tasks is large and the dimensionality of the space is limited). If we optimize $P^l_tP^l_q$ directly, we may get $P^l_t = 0$, or the column scale of $P^l_t$ is too small. To find a useful $P^l_t$, LcSP constrains the length of the column vector to be 1, i.e., to find $P^l_t$ on the Oblique Manifold ($\mathcal {OM}$). To reduce forgetting, we optimize the inter-task coherence $\mu(P^l_t,P^l_q)$ on $\mathcal{OM}$ so that the maximum entry of $P^l_tP^l_q$ is minimized.
> >
> > In the worst case, we may obtain $P^l_t$ consisting of **only one column vector** (i.e., $P^l_t$ with rank 1). To avoid this situation, we optimize the intra-task coherence $\mu(P^l_t)$ along with inter-task coherence, forcing the projector to satisfy the rank constraint in order to maintain the learning capacity of DNN (as shown in Fig.5 (a), the rank of projector affects the plasticity of DNN).
> >
> > In general, low-coherence projection, which can be seen as a relaxed orthogonal constraint, aims to balance plasticity and stability better, motivated by our observation that orthogonal projections reduce the plasticity of the model and make the DNN unable to adapt to new environments.

---

> > ### Author Response · Authors · 2022-11-18
> > **Response to Reviewer FMJV (Question 3-4)**
> >
> > **Question 3. The proposed method is limited to only the task-incremental setting where the task identifiers are required in both training and testing phases, as shown on page 3 and experiments. It further influences the significance of the work, considering the insignificant technical novelty, although this is with the same setting as (Chaudhry et al. 2020).**
> >
> > **Answer to Question 3.** Thank you for the comment. We further elaborate here on the **significance and insights** of this work to continual learning.
> >
> > This work considers a basic problem in continual learning, i.e., the capacity degradation problem in GOP methods. This work first explored this problem in the task-incremental setting and contributed an effective low-coherence subspace learning method, namely, LcSP, associated with its geometric optimization (optimization on Oblique manifold) and the theoretical analysis. The proposed solution is a general solution for most orthogonal projection-based methods. Therefore, the proposed LcSP is conveniently extended to the class incremental settings. For example, inferring its task label from the input data, which has been partially investigated in related work [2]. The insight of low-coherence subspace learning can also be used in other GOP methods to achieve a better Stability-Plasticity trade-off when learning long task sequences.
> >
> > **The further insights from this paper to continual learning.**
> >
> > **(1)** A common assumption in continual learning is that the number of tasks is unknown or infinite. However, most continual learning work evaluates the performance of algorithm on 20 tasks. This sequence of tasks may be so short as to obscure the shortcomings of the algorithm. In some cases, we need to evaluate the performance of algorithms on a long task sequence (e.g., Permuted CIFAR10 contains 64 tasks, and Permuted MNIST contains 150 tasks) and further develop algorithms that make more efficient use of network capacity. This work focuses on the continual learning on a long task sequence, a pending problem in the community.
> >
> > **(2)** It is difficult to make a balance between the plasticity and the stability of a model, which often influence each other. Memorizing all knowledge without forgetting may result in the model not being able to adapt to the new environment. An effective algorithm should also test whether it effectively balances plasticity and stability, i.e., it has the memory capacity while maintaining the learning capacity. This paper finds an efficient way to solve this difficult problem.
> >
> > [2] Nikhil Mehta, Kevin Liang, Vinay Kumar Verma, and Lawrence Carin. Continual learning using a bayesian nonparametric dictionary of weight factors. In International Conference on Artificial Intelligence and Statistics, pp. 100–108. PMLR, 2021.
> >
> >
> > **Question 4. Some detailed questions: -- What is the gap between the implementation and the analysis on page 6? -- How is the efficiency of the proposed method? How is the running time compared to other methods?**
> >
> > **Answer to Question 4.** We highly appreciate this valuable comment. **We have modified and added the efficiency analysis (the Table.4) in Appendix A.2 on page 15, which has been uploaded to this system.**
> >
> > Here are the modifications we have made in the Appendix.
> >
> > Based on the RTX3060 12G, we tested the efficiency of LcSP and compared methods on Permuted CIFAR10 (which contains 64 tasks, and each image's size is $3 \times 32\times32$), using AlexNet for OWM and GPM, and resnet18 for the other methods. We provide 4 metrics to assess the efficiency of the method: floating point operations **FLOPs**, time spent per epoch during training **Time (s)**, the number of epochs needed to train to convergence **Epochs**, and average time spent on inference **Mean_inference_time (ms)**. The detailed results are listed in Table 1 below.
> >
> > In addition, we calculated the total time taken by LcSP to create the projector for all tasks (64 in total): **3.211** seconds.
> >
> > **Table.1 Efficiency analysis results**
> >
> > |                         |    OWM     |    GPM     |  AdamNSCL   | ORTHOG-SUBSPACE |    LcSP     |
> > | ----------------------- | :--------: | :--------: | :---------: | :-------------: | :---------: |
> > | FLOPs                   | 26,676,256 | 23,202,880 | 558,538,752 |   556,708,864   | 528,965,376 |
> > | Time(s)      |   47.614   |   5.174    |   47.685    |     87.623      |   25.496    |
> > | Epochs                  |     25     |    200     |     80      |       80        |     80      |
> > | Mean_inference_time(ms) |   1.249    |   1.152    |   10.089    |      3.957      |    5.893    |

---

### Decision · Program_Chairs · 2023-01-20

**Decision:**

Reject

**Justification For Why Not Higher Score:**

There is some concern on the novelty of using low-coherence subspaces, but its application in continual learning appears new.  It is slightly surprising that the projection matrices are not specific to the tasks, and better clarifications are needed on why this still proffers good performance.  Empirically, the batch normalization seems to play an important role in keeping the good performance, and it will be important to understand its interaction with low-coherence.

**Justification For Why Not Lower Score:**

N/A

**Metareview: Summary, Strengths And Weaknesses:**

This paper addresses the issue of learning capacity degradation in gradient orthogonal projection methods that are used to combat catastrophic forgetting in continual learning.  The idea is to move from orthogonal subspaces to low-coherence subspaces, and a loss function is designed that accounts for gradient projections on the oblique manifold.  Experimental results show that the proposed approach can effectively combat forgetting while retaining good learning capacity, especially when there are many tasks.

 The paper addresses an important problem and is well written.  The use of low-coherence subspaces appears effective.  There is some concern on the novelty of using low-coherence subspaces, but its application in continual learning appears new.  It is slightly surprising that the projection matrices are not specific to the tasks, and better clarifications are needed on why this still proffers good performance.  Empirically, the batch normalization seems to play an important role in keeping the good performance, and it will be important to understand its interaction with low-coherence.


**Summary Of Ac-Reviewer Meeting:**

The reviewers and the AC met on Zoom for 40 minutes to discuss the paper.  While the paper does have some interesting insights that were never shown in continual learning (the effectiveness of low-coherence subspace), the reviewers are concerned about the experimental results, especially the interpretation and theoretical intuition, rather than just its performance (see the meta-review).  Although there are some concerns on novelty because it applies an existing approach to continual learning, this is considered a minor issue.  Overall, the weakness outweighed the merit, and the reviewers think the paper needs a further revision.